# Antiviral Mechanism of Virucidal Sialic Acid Modified Cyclodextrin

**DOI:** 10.3390/pharmaceutics15020582

**Published:** 2023-02-09

**Authors:** Yong Zhu, Andrey A. Sysoev, Paulo H. Jacob Silva, Marine Batista, Francesco Stellacci

**Affiliations:** 1Institute of Materials, École Polytechnique Fédérale de Lausanne, Switzerland, Station 12, 1015 Lausanne, Switzerland; 2Department of Mechanistic Cell Biology, Max-Planck Institute of Molecular Physiology, Otto-Hahn-Straße 11, 44227 Dortmund, Germany

**Keywords:** influenza, hemagglutinin, 6′SLN, hydrophobic linker, antiviral mechanism, virucidal, membrane interaction

## Abstract

We have reported that CD-6′SLN [6-sialyllactosamine (6′SLN)-modified β-cyclodextrin (CD)] can be a potential anti-influenza drug because it irreversibly deactivates virions. Indeed, in vivo, CD-6′SLN improved mice survival in an H1N1 infection model even when administered 24 h post-infection. Although CD-6′SLN was designed to target the viral envelope protein hemagglutinin (HA), a natural receptor of 6′SLN, it remains unclear whether other targets exist. In this study, we confirm that CD-6′SLN inhibits the influenza virus through an extracellular mechanism by interacting with HA, but not with neuraminidase (NA), despite the latter also having a binding pocket for the sialyl group. We find that CD-6′SLN interacts with the viral envelope as it elicits the release of a fluorophore embedded in the membrane. Two similar compounds were designed to test separately the effect of 6′SLN and of the undecyl moiety that links the CD to 6′SLN. Neither showed any interaction with the membrane nor the irreversible viral inhibition (virucidal), confirming that both components are essential to membrane interaction and virucidal action. Unlike similar antiviral cyclodextrins developed against other viruses, CD-6′SLN was not able to decapsulate viral RNA. Our findings support that combining viral protein-specific epitopes with hydrophobic linkers provides a strategy for developing antiviral drugs with a virucidal mechanism.

## 1. Introduction

Influenza infections affect 20–30% of the child population and up to 10% adult population globally and cause from 290,000 to 650,000 deaths every year [1], placing a burden on global health and the economy. Although anti-influenza drugs, such as oseltamivir, zanamivir, peramivir and baloxavir marboxil, have been approved for clinical use, the effectiveness of these drugs relies heavily on timely treatment, usually up to 48 h after the onset of symptoms [2]. Moreover, these drugs face challenges with drug resistance [3,4,5] due to the high mutation rate and frequent genetic reassortments of influenza viruses. Thus, developing novel influenza antivirals with longer therapeutical time windows and higher resistance to mutations is important to help in the treatment of this disease.

The influenza virion consists of a viral envelope and ribonucleoprotein complexes containing segmented, single-stranded RNA bound and encapsulated by nucleoproteins (Figure 1a). The outer envelope is a lipid bilayer with integrated two-spike glycoproteins, hemagglutinin (HA) and neuraminidase (NA), which are vital for mediating viral attachment and release, respectively. Under the lipid bilayer is the Matrix protein 1 (M1), which forms a capsid enclosing the genome. Matrix protein 2 (M2), a proton-selective ion channel necessary for viral genome release during virus entry, is integrated into the influenza outer envelope.

In general, influenza antiviral drugs target various stages of the viral replication cycle. Small-molecule viral enzyme inhibitors [2], such as the M2 proton channel blocker amantadine, endonuclease inhibitor baloxavir marboxil, and neuraminidase inhibitor oseltamivir, target the intracellular viral replication processes of viral fusion, genome replication, and viral budding, respectively. Large molecules, such as therapeutical antibodies [6], interact directly with extracellular virions, in particular viral membrane proteins, to prevent viral attachment or fusion to host cells. Notably, such an extracellular mechanism avoids many side effects, often associated with the impact on the intracellular physiology of host cells.

Influenza HA has been studied as a target of antiviral drugs for decades, though no drug candidate with an HA inhibition mechanism has been licensed for clinical use. HA binds to sialic acids of glycoproteins and glycolipids on the cell surface to allow viral entry into the host cell. Therefore, sialic acid-based glycomimetics could be a potential HA inhibitor. Various scaffolds such as glyco-nanoparticle [7], DNA [8], polyglycerol [9] and polyamidoamine dendrimer [10] modified with multiple sialic acid units showed enhanced hemagglutination inhibition or influenza inhibition compared with the monovalent sialic acids due to the multivalency effect [11,12]. Most of these studies explored how the valency and spacing affected the antiviral effect, while few focused on the role of the linker between the binding motif and the scaffold. 

In our previous study [13], we reported CD-6′SLN [6-sialyllactosamine (6′SLN)-modified β-cyclodextrin (CD)], a virucidal macromolecule made of β-cyclodextrin core covalently grafted with the sialyl trisaccharide Neu5Acα2-6Galβ1-4GlcNAc (6′SLN) via hydrophobic undecyl linkers (Figure 1b). β-cyclodextrin was selected as a scaffold because it’s a safe natural cyclic oligosaccharide with a well-defined structure and high biocompatibility [14,15,16]. We found CD-6′SLN showed nanomolar to micromolar inhibition against seasonal influenza strains H1N1, H3N2 and influenza B, while the analogous molecule with hydrophilic polyethylene glycol linkers CD-(Mal-PEG8)_7_-6′SLN (Figure 1c) only shows reversible inhibition with lower potency. In the BALB/c H1N1 mouse model, the groups treated with CD-6′SLN showed a much higher survival rate than the group treated with oseltamivir 24 h after infection. The mechanisms of viral inhibition and more specifically the mechanism of virucidal action of CD-6′SLN are not fully understood.

Here, we aim to elucidate the antiviral mechanism of CD-6′SLN. HA is most likely the main target of CD-6′SLN because 6′SLN is the natural binding moiety for HA in human strains such as H1N1 and H3N2 [17,18,19]. Moreover, HA is the most abundant envelope protein, with around 500 spikes per virion [20]. Although NA accounts for only 1/5th to 1/4th of the total spike proteins, it could also be a potential target as it also bears sialic acid binding pockets [21,22]. It is possible that the catalytic activity of NA is blocked by CD-6′SLN, causing inefficient viral entry [22,23], endocytosis and release [24]. Another potential target could be the viral envelope, as CD-6′SLN contains long hydrophobic chains that may interact with the viral envelope. CD-MUS [25] is a compound structurally similar to CD-6′SLN but carries sulfonate groups instead of 6′SLN at the end of the undecyl chains. It irreversibly inhibits many viruses that depend on heparan sulfate proteoglycan for cell entry. Hence, both CD-6′SLN and CD-MUS are effective virucidal antivirals. While CD-MUS was designed to target the viral attachment ligands (VALs), it has been shown to disrupt the structural integrity of herpes simplex virus-2, forcing the release of the viral genomic DNA. A similar molecular mechanism may also contribute to the virucidal activity of CD-6′SLN.

We investigated the molecular targets of CD-6′SLN by applying the compound during various stages of infection. In addition, we performed assays to confirm whether viral membrane proteins and envelope interact with the compound and if the compound was able to break down the virus to release its genome. Our findings demonstrate that CD-6′SLN targets hemagglutinin and affects the viral lipid membrane. Unlike CD-MUS, CD-6′SLN did not cause the release of the viral genome, suggesting a somewhat different mechanism from that reported for CD-MUS [25].

## 2. Materials and Methods

### 2.1. Cells, Virus and Chemicals

MDCK (Madin-Darby Canine Kidney Cells) were purchased from American Type Culture Collection (ATCC, Manassas, VA, USA). The cells were cultured in Dulbecco’s modified Eagle’s medium (DMEM, Thermo Fisher Scientific, Waltham, MA, USA) with 1% Penicillin/Streptomycin and 10% FBS (Thermo Fisher Scientific, Waltham, MA, USA) in the humidified atmosphere in 5% CO_2_ at 37 °C.

H1N1 (A/Netherlands/2009) was a kind gift from Prof. M Schmolke (University of Geneva) and it was reproduced in MDCK cells and titrated by immunocytochemistry assay.

Heptakis-(6-deoxy-6-iodo)-β-Cyclodextrin was purchased from AraChem (Tilburg, Netherlands) and dried under high vacuum for 48 h prior to use. 12-mercaptododecanoic acid was purchased from Sigma-Aldrich (St. Louis, MO, USA) and dried in the same way. Neu5Acα2-6Galβ1-4GlcNAc-β-ethylamine was purchased from Asparia Glycomics (Donostia, Spain). Maleimide-PEG8-CH_2_CH_2_COOH was purchased from PurePEG (San Diego, CA, USA). All the other chemicals were purchased from Sigma-Aldrich (St. Louis, MO, USA) as reagent grade and used without further purification. 

### 2.2. Chemical Synthesis and Characterization

CD-(S-C11-COOH)_7_, CD-6′SLN, and CD-(Mal-PEG8)_7_-6′SLN were synthesized based on a previously published procedure [13], with modifications as described below. The results of NMR and mass spec of the molecules are shown in Appendix A.

#### 2.2.1. CD-(S-C11-COOH)_7_

Potassium tert-butoxide (48.51 mmol, 5.443 g) and 12-mercaptododecanoic acid (22.05 mmol: 5.123 g) were dissolved in 50 mL of DMF under an argon atmosphere. The viscous mixture was stirred at room temperature for 30 min using a mechanical stirrer. Heptakis-(6-deoxy-6-iodo)-β-Cyclodextrin (2.625 mmol, 5 g) was dissolved separately in dry DMF and added to the mixture. The mixture was placed in an oil bath at 70 °C and allowed to react for 12 h. The crude was precipitated directly into 400 mL diethyl ether (Et_2_O) and vacuum filtered with a fritted disk funnel (Por 3). The solid on the filter was collected (5 g), dried under vacuum for 24 h, and then dissolved in a solution of 95 mL distilled water. This solution was precipitated into 190 mL Acetonitrile and filtered on a fritted disk (Por 4). The material was collected, dissolved in MilliQ water, and freeze-dried to obtain 1.05 g CD-(S-C11-COOH)_7_. ^1^H NMR (400 MHz, TFA-*d*) δ 5.37 (d, *J* = 3.6 Hz, 1H, glucose H-1), 4.35 (t, *J* = 9.0 Hz, 1H, glucose H-3), 4.28 (t, *J* = 7.4 Hz, 1H, glucose H-5), 4.10 (dd, *J* = 9.9, 3.4 Hz, 1H, glucose H-2), 3.88 (t, *J* = 9.3 Hz, 1H, glucose H-4), 3.48 (d, *J* = 12.8 Hz, 1H, glucose H-6a), 3.24 (dd, *J* = 13.5, 7.6 Hz, 1H, glucose H-6b), 3.00–2.84 (m, 2H, S-CH_2_-CH_2_), 2.64 (t, *J* = 7.6 Hz, 2H, CH_2_-COOH), 1.86 (q, *J* = 7.4 Hz, 4H, CH_2_-CH_2_-COOH, S-CH_2_-CH_2_), 1.53 (m, 14H, S-C-C-CH_2_-CH_2_-CH_2_-CH_2_-CH_2_-CH_2_-CH_2_-C-C-COO). HRMS (nanochip-ESI/LTQ-Orbitrap) *m*/*z*: [M–3H]^3−^ Calcd for C_126_H_221_O_42_S_7_^3−^ 877.5133; Found 877.1148.

#### 2.2.2. CD-6′SLN

1.05 g CD-(S-C11-COOH)_7_ was dissolved in 15 mL DMSO with the addition of N-hydroxysuccinimide (5.51 mmol, 634 mg), 1-Ethyl-3-(3-dimethylaminopropyl) carbodiimide hydrochloride (10.2 mmol, 2 g), and 4-Dimethylaminopyridine (0.486 mmol, 60 mg) and stirred overnight. The crude sample was washed with 200 µL acidic water (1 M HCl in H_2_O) and centrifuged at 5500× *g* for 1 min. This step was repeated two more times followed by centrifugations at 5500× *g* for 10 min and the pellet was collected. A solution of 50:50 ACN: Et_2_O (15 mL) was added to the pellet and centrifuged again at 5500× *g* for 5 min after the resuspension of the pellet. The new pellet was collected, followed by the addition of Et_2_O (15 mL) and centrifugation at 5500× *g* for 5 min. The final pellet was collected and vacuum dried. A portion of the collected activated cyclodextrin derivative (0.28 mmol: 935 mg) and the Neu5Acα2-6Galβ1-4GlcNAc-β-ethylamine (6′SLN, 0.63 mmol: 467 mg) were dissolved in 16 mL DMSO. 100 µL of a triethylamine solution (200 µL TEA in 1 mL DMSO) was added to the DMSO solution and the mixture was stirred overnight. The next day, the resulting material was collected and was dialysed against MilliQ water for three days (the dialysis water was changed twice per day) in a cellulose membrane (MWCO: 2kDa). The material left in the membrane was collected at the end of three days and lyophilized. ^1^H NMR (400 MHz, D_2_O) δ 5.13 (s, 2.8H, β-CD Glc H-1) 4.58 (s, 1H, GlcNAc H-1), 4.46 (d, *J* = 7.8 Hz, 1H, Gal H-1), 4.16–3.29 (m, 31H, other H on 6′SLN and Glc H, S-CH_2_-CH_2_), 2.68 (d, *J* = 11.9 Hz, 1H, Neu5Ac H-2), 2.29 (m, 2.2H, CH_2_COO), 2.06 (d, *J* = 9.8 Hz, 6H, COCH_3_), 1.82–1.11 (m, 19H, S-C-CH_2_-CH_2_-CH_2_-CH_2_-CH_2_-CH_2_-CH_2_-CH_2_-CH_2_-C-COO, Neu5Ac H-2′).

#### 2.2.3. CD-(Mal-PEG8)_7_-6′SLN

Heptakis-(6-deoxy-6-mercapto)-β-Cyclodextrin (0.024 mmol, 30 mg) was dissolved into 1.5 mL DMSO and mixed with 8 mL 0.1 M phosphate buffer (pH 6.8) containing Maleimide-PEG8-CH_2_CH_2_COOH (0.29 mmol, 150 mg). The reaction mixture was stirred overnight at room temperature and then dialysed against MiliQ water with cellulose membrane (MWCO: 1kDa) for three days and dried by lyophilization to yield pink powder CD-(Mal-PEG8)_7_-COOH. HRMS (nanochip-ESI/LTQ-Orbitrap) m/z: [M–4H]^4−^ Calcd for C_203_H_339_N_7_O_112_S_7_^4−^1223.5775; Found 1223.2257. CD-(Mal-PEG8)_7_-COOH (15.6 µmol, 76.4 mg), N-hydroxysuccinimide (0.42 mmol, 48 mg), 1-Ethyl-3-(3-dimethylaminopropyl) carbodiimide hydrochloride (0.18 mmol, 36 mg) and 4-Dimethylaminopyridine (0.0097 mmol, 1.2 mg) were dissolved in 5 mL DMSO and stirred overnight. After reaction, the product was precipitated and washed by 500 mL dichloromethane/diethyl ether (1/50) mixture twice and dried under vacuum. 10 mg of obtained cyclodextrin intermediate was dissolved in 2 mL DMSO. Neu5Acα2-6Galβ1-4GlcNAc-β-ethylamine (4.6 µmol, 3.3 mg) and triethylamine (0.049 mmol, 5 mg) were then added in. The mixture was stirred overnight and the product CD-(Mal-PEG8)_7_-6′SLN was purified by three-day dialysis against MiliQ water with cellulose membrane (MWCO: 2kDa) and dried by lyophilization.

### 2.3. H1N1 Inhibition Assay

MDCK cells were pre-seeded on a 96-well plate at the density of 2 × 10^4^ cells per well 24 h in advance. 0.005 MOI of H1N1 was used for all inhibition experiments. For the virus pretreatment groups, 10 μg/mL CD-6′SLN were incubated with H1N1 at 37 °C for 1 h and the mixture was added to the cell cultures. For the cell pretreatment groups, the cells were first treated with 10 μg/mL CD-6′SLN for 1 h and then infected with the virus. For the virus co-treatment groups, 10 μg/mL CD-6′SLN were added simultaneously with the virus to the cells. For the post-infection treatment groups, the virus was added to the cells, after 1 h incubation for viral attachment, the inoculum was removed and a medium containing 10 μg/mL CD-6′SLN was added to the cells. After 24 h incubation, the medium was removed and cells were fixed with methanol for 1 h. Then the LIGHT DIAGNOSTICS Flu A monoclonal antibody (1:100 dilution, Merck Millipore, Burlington, MA, USA) was added and incubated for 1 h at 37 °C. The cells were washed with wash buffer (PBS + Tween 0.05%) three times, followed by adding anti-mouse lgG, HRP-linked antibody (1:500 dilution, Cell Signaling Technology, Danvers, MA, USA). After 1 h, the cells were washed and the DAB solution was added. Infected cells were counted and percentages of infection were calculated by comparing the number of infected cells in treated and untreated conditions.

The EC_50_ measurements of CD-6′SLN, CD-(S-C11-COOH)_7_, and CD-(Mal-PEG8)7-6′SLN were performed using the virus pretreatment method. Prism 9 (GraphPad, San Diego, CA, USA) was used to estimate the EC_50_ from the dose-response curve (Appendix A).

### 2.4. Biolayer Interferometry 

Gator^TM^ Anti-His probes (No. 160009, Gator bio, Palo Alto, CA, USA) were first prewetted in 10 mM PBS buffer (75 mM NaCl, pH 7.6), and then immobilized with 10 μg/mL H1N1 Hemagglutinin (A/California/04/2009, GSC-Z03181, GenScript, Rijswijk, The Netherlands) for 300 s to reach a shift of 1.8 nm. After balancing the probes in the same buffer for 60 s, the probes were dipped into different concentrations of the test materials for 300 s, and then into the buffer for 900 s. Reference probes for each concentration were undergoing the same procedure without HA immobilization. The association and dissociation shift (nm) were plotted by subtracting the signal of reference probes from experimental probes, and the equilibrium dissociation constant K_D_ was calculated by steady-state analysis using Prism 9 (GraphPad, San Diego, CA, USA).

### 2.5. Neuraminidase Inhibition Assay 

The activities of Influenza H1N1 (A/California/04/2009) Neuraminidase (11058-VNAHC, Sino Biological, Bejing, China) were tested using a fluorogenic substrate MUNANA (2′-(4-Methylumbelliferyl)-alpha-D-N-acetylneuraminic acid sodium salt hydrate, Sigma Aldrich, 69587). The optimal final concentration of NA (1.75 unit/mL) and MUNANA (100 μM) were chosen after pretests to ensure the linear relationship between RFU readout and the fluorescence product concentration. In the assay, NA was incubated with different concentrations of tested materials in 33 mM MES buffer (4 mM CaCl_2_, pH 6.5) for 2 h. MUNANA was then added to the mixture and the fluorescence was measured after 1 h incubation at 37 °C using a Tecan Spark plate reader (excitation wavelength 355 ± 30 nm, emission wavelength 460 ± 25 nm). Prism 9 (GraphPad, San Diego, CA, USA) was used to determine the IC_50_ from the dose-response curve.

### 2.6. RNA Exposure Assay

Influenza H1N1 (A/Netherlands/2009, 1.3 × 10^5^ PFU/mL) was incubated with 100 μg/mL tested material in DMEM medium containing 1% P/S at 37 °C for 3 h. 1.1 μL 10 mg/mL RNAse A (EN0531, Thermo Fisher Scientific, Waltham, MA, USA) or 1.1 μL MiliQ water was added into 10 μL 1× PBS buffer, 100 μL 1:20 diluted viral material solution in two Eppendorf tubes. After 30 min incubation at 37 °C, 500 μL GTC lysis buffer (Omega bio-tek, Norcross, GA, USA) was added into each tube and RNA was extracted using E.Z.N.A. DNA/RNA/isolation kit (Omega Bio-tek, Norcross, GA, USA) and quantified by RP-qPCR. The RT-qPCR reaction (final volume 10 μL) consisted of 5 μL QuantiTech Probe RT-PCR Master Mix 2×, 0.1 μL QuantiTect RT Mix, 2.5 μL extracted viral RNA and 2.4 μL primers (4 μM each, 5’-AAG ACC AAT CYT GTC ACC TCT GA-3’, 5’-CAA AGC GTC TAC GCT GCA GTC C-3’, Microsynth AG, Balgach, Switzerland) and probe (2 μM, FAM5’-TTT GTG TTC ACG CTC ACC GTG CC-3’TAMRA, Microsynth AG, Balgach, Switzerland). The RT-qPCR was executed on QuantiStudio 7 (Thermo Fisher Scientific, Waltham, MA, USA), starting at 50 °C for 30 min and 95 °C for 15 min, followed by 45 cycles of 15 s at 94 °C and 1 min at 60 °C. Cycle threshold (Ct) was analysed by QuantiStudio software v1.7.2(Thermo Fisher Scientific, Waltham, MA, USA) and ∆Ct was plotted as the difference between the RNAse-treated sample and MilliQ water-treated sample.

### 2.7. R18-Labeled H1N1 Preparation

2.33 μL of 3.12 μmol/mL ethanolic R18 (octadecyl rhodamine B chloride) solution was injected into 255 μL H1N1 suspension containing 2 mg/mL of viral protein in total, quantified by modified Lowry protein assay Thermo Scientific™ Pierce™ Modified Lowry Protein Assay Kit (Thermo Fisher Scientific, Waltham, MA, USA) under vortex mixing. The mixture was incubated in the dark at room temperature for 1 h, and the non-inserted fluorophore was removed by chromatography on Zeba™ Spin Desalting column (7K MWCO, Thermo Fisher Scientific, Waltham, MA, USA) using 10 mM TES, 150 mM NaCl (pH 7.4) as elution buffer. The protein concentration of the R18-labeled virus was determined again by the modified Lowry assay.

### 2.8. R18 Release Assay

R18-labelled H1N1 (final viral protein concentration 2 μg/mL) was added into 1% P/S DMEM medium containing triton X-100, CD-6′SLN, CD-(S-C11-COOH)_7_ or CD-(Mal-PEG8)_7_-6′SLN. Fluorescence was measured at different timepoints using Tecan infinite 200 plate reader with excitation at 560 nm and emission at 590 nm. 

### 2.9. H1N1 Virucidal Assay

100 μg/mL test compound was incubated with 1.3 × 10^5^ FFU/mL H1N1 at 37 °C. After 15 min, 30 min, 45 min, 1 h, 1.5 h, 2 h, 2.5 h and 3 h, the virus compound mixture was titrated using the ICC assay mentioned in the H1N1 inhibition assay. Viral titre calculated the number of stained cells times the dilution factor.

## 3. Results

### 3.1. CD-6′SLN Inhibits H1N1 Virus Primarily through an Extracellular Mechanism

We first tested the anti-influenza efficacy of CD-6′SLN with different treatments. The infection of influenza starts with viral attachment, which usually takes 1 h on Madin-Darby canine kidney (MDCK) cells. Subsequently, the viral genome replicates in the cell and viral proteins are produced in the next 24 h. In this experiment, the cells and virus were treated with various concentrations of CD-6′SLN under one of the following four different conditions: (i)cell pretreatment: CD-6′SLN preincubation with the cells 1 h before cell inoculation with the virus;(ii)virus pretreatment: CD-6′SLN addition to the virus inoculum 1 h before addition to the cells;(iii)cotreatment: simultaneous addition of CD-6′SLN and of the viral inoculum to the cells;(iv)post-treatment: CD-6′SLN addition after viral attachment.

The infection percentage was measured by immunocytochemistry (ICC) assay. The strongest inhibition occurred under virus pretreatment [EC_50_ = 0.025 μg/mL (95% CI 0.017–0.035 μg/mL)] followed by cotreatment [EC_50_ = 0.19 μg/mL (95% CI 0.11–0.32 μg/mL)]. Cell pretreatment and post-treatment showed some inhibition but only at a high concentration of 100 μg/mL (all results are shown in Figure 2). In comparison, cell pretreatment and post-treatment showed inhibition only at a high concentration of 100 μg/mL. The diminished antiviral effect observed after viral entry (post-treatment vs. virus pretreatment) indicates that CD-6′SLN inhibits the early stage of viral infection. Meanwhile, treating the cells in the absence of the virus also showed a weaker antiviral effect (cell pretreatment vs. virus pretreatment), confirming that CD-6′SLN acts on the virus rather than host cells. Therefore, we conclude that CD-6′SLN most likely acts directly on viral particles to block viral attachment, instead of preventing viral replication inside host cells.

### 3.2. CD-6′SLN Binds to HA but Not NA

To confirm HA that is the target protein for CD-6′SLN, we used biolayer interferometry to study the binding affinity between CD-6′SLN and recombinant HA (A/California/04/2009, Met1-Gln549) immobilized with His-Tag. The protein was coupled to probes precoated with His-Tag antibody; the probes were dipped into CD-6′SLN solutions, and then washed to record the association and dissociation processes. We observed dose-dependent interaction between CD-6′SLN and HA (Figure 3a), with an estimated binding affinity of 8.79 μM [95% confidence interval (CI) 6.61–11.68 μM]. By contrast, the trisaccharide ligand 6′SLN itself did not show any measurable binding to HA up to 5000 μM, neither did the non-6′SLN modified scaffold CD-(S-C11-COOH)_7_ (structure shown in Figure 1c) at 200 μM. This confirmed that our strategy of conjugating multiple 6′SLN ligands in close proximity in the CD-6′SLN macromolecule led to significantly stronger binding towards HA. Given that each CD-6′SLN macromolecule contained 2.5 6′SLN ligands on average, we estimate that the affinity of each 6′SLN ligand in CD-6′SLN towards HA is at least 227-fold higher compared to free 6′SLN because of multivalency. 

The interaction between CD-6′SLN and NA was studied using an enzyme inhibition assay. A fluorophore generative substrate, sodium (4-methylumbelliferyl-α-d-N-acetylneuraminate), was used to probe the catalytic activity of NA (A/California/04/2009) pre-treated with CD-6′SLN for 1 h. The progression of the reaction was monitored by measuring the fluorescence emission of the reaction mixture after 1 h incubation. Oseltamivir, a known NA inhibitor, was used as a positive control and showed nanomolar inhibition against NA (Figure 3b). Meanwhile, CD-6′SLN showed inhibition only at significantly high concentrations [IC_50_ = 0.53 mM (95% CI 0.43–0.69 mM)]. Altogether, the data from biolayer interferometry and enzyme inhibition assay indicate that the antiviral activity of CD-6′SLN likely relies on the interaction with hemagglutinin and not with neuraminidase.

### 3.3. CD-6′SLN Does Not Release Viral Genome 

We sought to confirm the disruption of the viral envelope using the RNA exposure assay [25], which detects whether viral genomic RNA, normally shielded by the envelope, could be released upon the addition of the compound. Briefly, the mixture of H1N1 and CD-6′SLN was treated with RNase A or buffer, and RNA in each sample was extracted and quantified with RT-qPCR (Figure 4a). RNase could degrade the free viral RNA released by the compound but not the genomic RNA protected by viral structural proteins and envelope [25]. Therefore, by comparing the difference in the cycle threshold between the RNase-treated and buffer-treated samples, we are able to determine whether the viral genome is released or not. Unexpectedly, we found that CD-6′SLN, unlike sodium dodecyl sulfate (SDS) or CD-MUS [25], did not show any significant change in the detectable RNA content compared to the untreated control (Figure 4b). We reason that this could be due to the incomplete disruption of viral structure. As it is illustrated in Figure 1b, the viral RNA is encapsulated by nucleoproteins and also protected by a layer of matrix protein and viral envelope. It is possible that CD-6′SLN only disrupted the structure of the viral envelope after binding specifically to HA, while the matrix protein layer and nucleoprotein encapsulation remained intact, making the RNA inaccessible to the enzyme.

### 3.4. CD-6′SLN Affects Viral Envelope

To further investigate if the viral envelope was affected during incubation with CD-6′SLN, we labelled the viral envelope with octadecyl rhodamine B (commonly known as R18), a hydrophobic dye that is quenched upon incorporation into a lipid membrane. This way, the fluorescence signal appears when there is membrane disruption [26]. By measuring the dynamic fluorescence release of the virus incubated with CD-6′SLN, we were able to monitor its interaction with the viral envelope in real-time. Triton X-100 was used as a positive control as it had been widely used for damaging the viral envelope in similar assays [27]. The fluorescence signal from viruses treated with CD-6′SLN gradually increased in the first 40 min, then after some fluctuations, it reached a plateau of 1200 RFU after 2 h. The signal generated by the addition of Triton X-100 increased in the first 20 min and then slowly decreased reaching a plateau at 500 RFU after 2 h (Figure 5a). Importantly the initial increase in signal for the case of Triton X-100 is much more rapid than in the case of CD-6′SLN. These results show that CD-6′SLN interacts with and probably disrupts the viral envelope with a mechanism kinetically different from that of Triton X-100. In contrast, CD-(S-C11-COOH)_7_ did not cause any change in the fluorescence signal, hence it does not elicit the release of R18 from the viral envelope, consistently with its inability to inhibit H1N1 in vitro (Appendix A). In addition, CD-(Mal-PEG8)_7_-6′SLN (Figure 5c), a compound that differs from CD-6′SLN only in its hydrophilic linker and that has been shown to be virustatic against H1N1 [reversible inhibition, EC_50_ = 1.86 μg/mL (95% CI 1.05–3.42 μg/mL), Appendix A, showed no changes in the fluorescence signal when mixed with R18 labelled H1N1. We interpret this result as an indication that CD-(Mal-PEG8)_7_-6′SLN does not damage the viral envelope. Taken all together, these results suggest that the ability to interact with the viral envelope is key to achieving irreversible viral inhibition. 

To investigate if the fluorescence assay described above is a good indication for the irreversible inhibition of the virus, we compared the kinetics of the process measured through the assay with the kinetics of irreversible inhibition acquired with in vitro virucidal assays at different timepoints. For the latter assays, we performed the tests after incubating the virus and CD-6′SLN together for 15, 30, 45, 90, 120, 150 and 180 min. We report in Figure 5b the resulting viral titres after sequential dilution. We found that the viral titre kept decreasing until 60 min incubation, showing the irreversible effect that CD-6′SLN has on H1N1 requires a full hour to be complete. We note that the fluorescence signal kept increasing for the first 70 min of the experiment. Overall, these kinetic trends are comparable, indicating that the fluorophore release is a good method to investigate the virucidal activity of these compounds.

## 4. Discussion and Conclusions

Most approved small-molecule antivirals inhibit the intracellular replication process, which may induce cell mutagenesis [28], carcinogenesis [29], and teratogenesis [30] depending on the specific mechanism. Therefore, developing new safe antivirals, focusing on novel mechanisms is essential to, not only increase the antiviral toolkit but also be ready for emerging viral diseases. Here we showed that CD-6′SLN inhibited influenza A H1N1 with an extracellular mechanism by interacting with the viral envelope protein hemagglutinin and viral envelope lipids that ultimately blocked viral entry. In this case, both the hemagglutinin ligand 6′SLN and aliphatic linker contribute to the antiviral effect. The 6′SLN group directly binds to HA on the viral envelope and CD-6′SLN interacts with the viral envelope thanks to its hydrophobic linker, likely causing envelope destruction. 

The prevalence and accessibility of hemagglutinin make it an important drug discovery target. Nevertheless, small sialic acid-containing inhibitors are not effective in this case due to the low affinity and mutational variability of the sialic acid binding site [31]. It seems that CD-6′SLN overcomes this limitation due to the multivalent interaction and irreversible damage mediated by the hydrophobic linker. Meanwhile, the natural ligand mimic provides a simple specificity mechanism, and the modular construction of CD-6′SLN enables further modifications in response to mutational variations of the sialic acid binding site.

CD-6′SLN is not the only antiviral that targets lipid bilayers. N-docosanol, which has a similar saturated fatty structure of 22 carbon atoms, is a commercial anti-herpes simplex virus (HSV-2) drug and has been reported to inhibit enveloped viruses including HSV-1, HSV-2, respiratory syncytial virus, cytomegalovirus, varicella-zoster virus, human herpesvirus 6 and human immunodeficiency virus-1 with EC_50_ ranging from 3 mM to 12 mM [32]. Unlike CD-6′SLN, n-docosanol is not directly virucidal and the optimal viral inhibition occurs when the compound is applied to host cells prior to viral infection. The underlying mechanism is that n-docosanol incorporates into cell membranes and is metabolized into phosphatidylcholine- and phosphatidylethanolamine-like species [33], thus, inhibiting viral fusion in the early stage of HSV-2 replication [34]. The inhibition activity of CD-6′SLN based on its action on the virus instead of the host cells, and the resulting virucidal effect, could probably be explained by the enrichment of CD-6′SLN on the viral envelope induced by specific 6SLN-hemagglutinin interactions.

In summary, the influenza inhibition and virucidal activity of CD-6′SLN result from the compound binding to the viral protein hemagglutinin and interacting with the viral envelope. This result implies that combining a virus-specific epitope with a lipid-targeting aliphatic linker is likely a general strategy for developing virucidal yet non-toxic antivirals. Based on the dual action of the viral protein and the envelope, we hypothesize that such compounds may be also effective against emergent viral strains in the future.

## Figures and Tables

**Figure 1 pharmaceutics-15-00582-f001:**
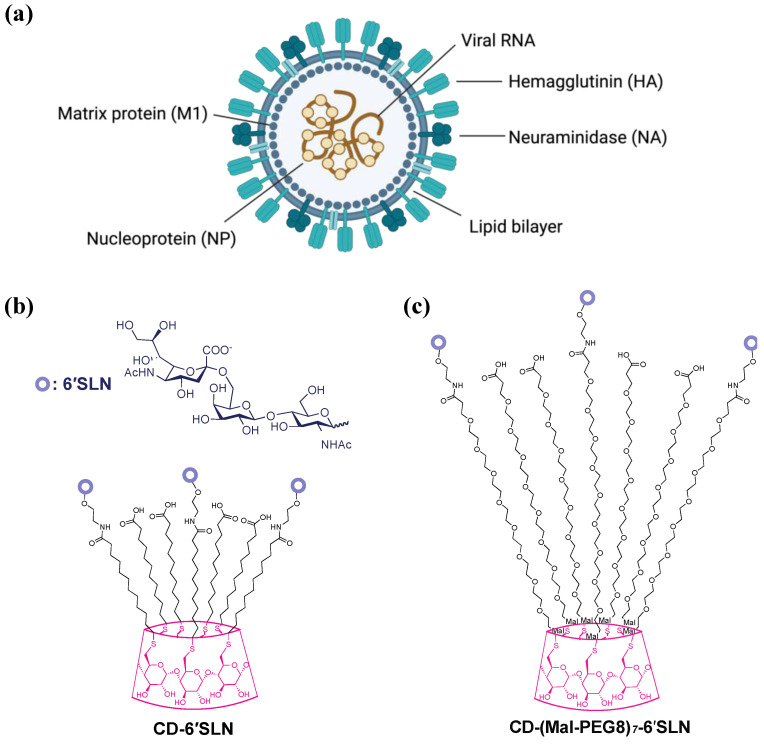
(**a**) Schematic drawing of the influenza virus. (**b**) Chemical structure of virucidal CD-6′SLN; schematic drawing of the influenza virus. (**c**) Chemical structure of CD-(Mal-PEG8)_7_-6′SLN.

**Figure 2 pharmaceutics-15-00582-f002:**
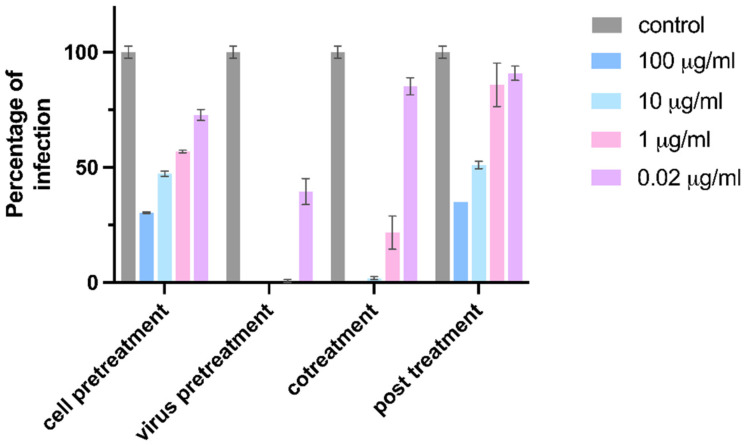
CD-6′SLN inhibition of H1N1 during various stages of viral infection. Cell pretreatment: compound was applied to the cells for 1 h, and the virus was added after washing the cell. Virus pretreatment: the virus was incubated with the compound for 1 h and added to the cells. Cotreatment: virus and compound were added simultaneously to the cells. Post-treatment: compound was added after viral attachment.

**Figure 3 pharmaceutics-15-00582-f003:**
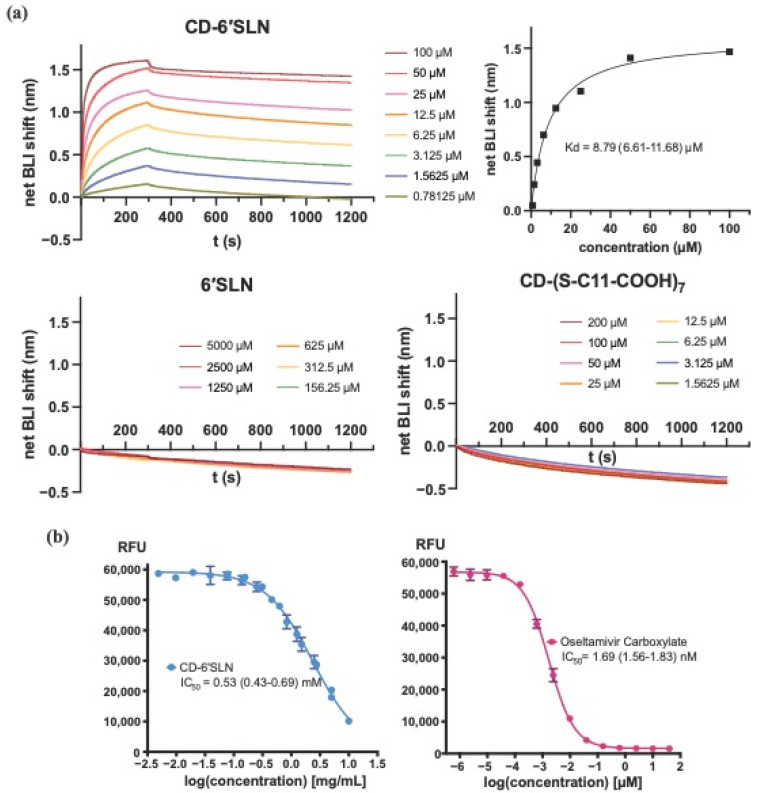
CD-6′SLN bind to H1N1 hemagglutinin instead of neuraminidase. (**a**) Binding of CD-6′SLN, 6′SLN and CD-(S-C11-COOH)_7_ to immobilized his-tagged hemagglutinin (H1N1, A/California/04/2009). (**b**) Dose-response inhibition of neuraminidase (H1N1, A/California/04/2009) catalytic activity by CD-6′SLN and oseltamivir carboxylate.

**Figure 4 pharmaceutics-15-00582-f004:**
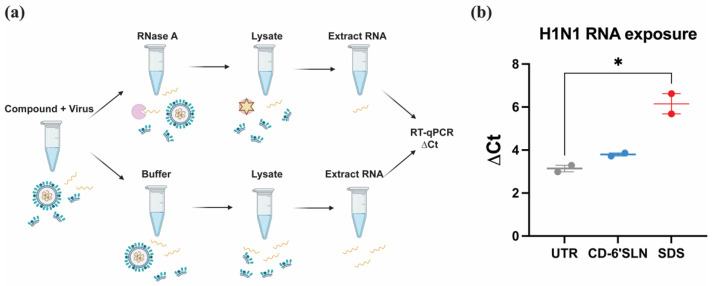
RNA exposure assay (**a**) Schematic drawing demonstrating the mechanism of the RNA exposure assay. The compound and virus were co-incubated for 3 h and then treated either with RNAse or buffer, followed by viral lysis. The RNA quantities in the lysate were quantified by RT-qPCR. (**b**) Cycle threshold difference (ΔCt = Ct_RNase_ − Ct_control_) of H1N1 samples treated with 100 μg/mL CD-6′SLN and 100 μg/mL SDS. Results are expressed as average ± standard deviation based on two independent experiments. The asterisk represents the *p*-value (*, <0.05) calculated by a two-tailed unpaired *t*-test.

**Figure 5 pharmaceutics-15-00582-f005:**
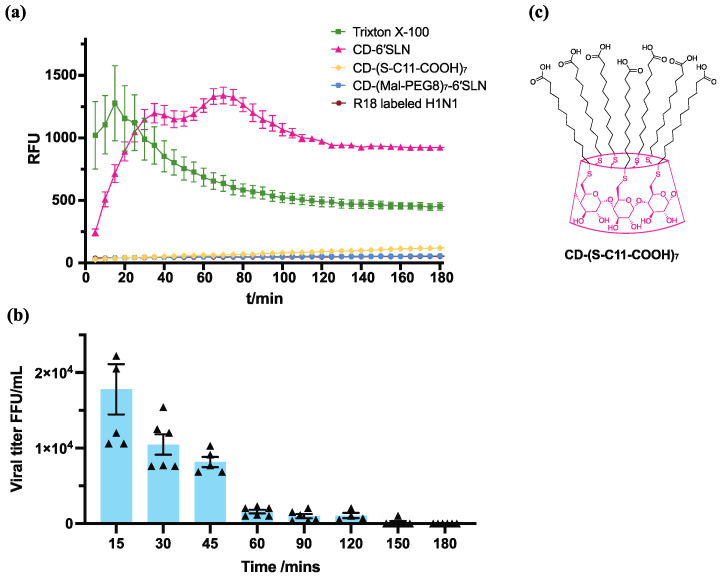
CD-6′SLN interacts with the viral envelope. (**a**) Fluorescence intensity of R18-labeled H1N1 treated with 100 μg/mL CD-6′SLN, 100 μg/mL CD-(S-C11-COOH)_7_, 100 μg/mL CD-(Mal-PEG8)_7_-6′SLN or 1% triton X-100. (**b**) Infectious H1N1 titre of the virus-CD-6′SLN mixture over time. Every triangle represents the viral titre determined in a single well. (**c**) Chemical structure of CD-(S-C11-COOH)_7_.

## Data Availability

The data presented in this study are openly available in FigShare at https://doi.org/10.6084/m9.figshare.21975689.v1 (accessed on 20 January 2023).

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
