# Peer review of "Antiviral Mechanism of Virucidal Sialic Acid Modified Cyclodextrin"

_pharmaceutics, 2023, doi:10.3390/pharmaceutics15020582_

Round 1

Reviewer 1 Report

Pharmaceutics_2022_2094632 The manuscript entitled “Antiviral mechanism of virucidal sialic acid modified cyclodextrin
by Yong Zhu, Andrey A. Sysoev, Paulo Jacob Silva, Marine Batista, and Francesco Stellacci.

The manuscript represents the research article and is devoted to beta-cyclodextrin-modified sialyllactosamine with undecyl linkers as a potential anti-influenza agent.

The authors carried out a very serious and thorough research, which seems important for the development of drug candidates against the influenza virus.

Along with biological tests (H1N1 inhibition assay, neuraminidase inhibition assay, RNA exposure assay, and others), the synthetic part is well represented, which is commendable.

The authors logically substantiated the choice of sialic acid and the length of the linker. I would like to understand what prompted the authors to modify the cyclodextrin skeleton. This explanation should be added to the manuscript.

These materials could be useful to specialists in the field of medicinal chemistry and pharmacology. They will also be of interest to biochemists and chemists working with bioactive compounds.

The manuscript does not raise any objections, and can be published in Pharmaceutics after minor revision.

Several details and inaccuracies should be noted.

1.    Introduction. Lines 78-82. Concerning CD-MUS106. There is no clarity, the essence is vague.

2.    Figures. On Fig.1a, it is desirable to draw chemical formulas larger and more clearly, especially the trisaccharide part, as in the cited work [7].

3.  Materials and Methods. Line 144. 1-Ethyl-3-(3-dimethylaminopropyl) carbodiimide (0.18 mmol, 36 g). Should be corrected.

4.    Results. Lines 214-223. It is necessary to describe in more detail the results of the processes at each of the four stages of viral infection, so that the conclusions made by the authors are substantiated.

Reviewer 2 Report

The manuscript by Stellacci et al describes/confirm that CD-6'SLN inhibited anti-influenza virus activity with an extracellular mechanism by targeting hemagglutinin (HA) and viral envelope. The synthesis reported is achieved according to previously published methods. Compared to previously published work on Adv. Sci. 2021, 8, 20011012 (ref 7), the present word added some additional proof to support the published results. And a lot of related papers based on multivalent 6′-sialyllalctosamine derivatives (eg. Carbohydr. Polym. 2016 Nov 20; 153: 96; Bioconjug Chem. 2018 May 16; 29(5): 1490, J. Med. Chem. 2021, 64, 12774, etc) showed antiviral activity by targeting HA protein. It is not surprising the virucidal activity of CD-6'SLN attribute to the binding with hemagglutinin. Therefore, the present manuscript fails to provide meaningful insights and is a rather limited piece of work. The authors are going to need a more complicated mechanism study to explain the antiviral results reported. Based on the amounts of work from this manuscript and significance of added information, the manuscript submitted is not ready for publication.

There are other issues with this manuscript which ought to be fully addressed prior to submit somewhere else.

The authors should include a paragraph about the use of 6'SLN building blocks for HA inhibitors citing all the relevant work, not just that from the authors lab.

For the biolayer interferometry assay, (i) another control—the interaction of CD-(S-C11-COOH)7 with HA protein should also be carried out. (ii) In this assay, the author only set four concentrations to obtain the binding affinity. More concentrations should be designed. (iii) line 238, the description of the sentance“an over 50-fold increase in affinity towards HA is achieved.” is not clearly described. In fact, the average number of 6’SLN is 2.5 per beta-CD in CD-6'SLN, the number 50 should further divided by 2.5?

The manuscript contains a number of spelling mistakes, grammatical errors and word use errors. The authors should double check it. For example:

Line 49-50, “Small-molecule viral enzyme inhibitors, such as M2 proton channel inhibitor amantadine,……”M2 proton channel protein not enzyme.

Line 56-57, Figure 1, (1a) the chemical structure of the sugars are wrong; (1b) the word “hemagglutin (HA)”is wrong;

Line 65, “glycol linkers CD-(Mal-PEG8)7-6’SLN (Figure 4c) only”, no figure 4c was found in the manuscript.

Line 119, “4.31 (dt, J = 29.7, 9.5 Hz, 2H, glucose H-3, H-5)”, can the author explain the J constant of 29.7 Hz?

Line 122, “1.53 (d, J = 16.1 Hz, 14H, other alkyl H)“???

Mixed use of formats, such as beta vs β; CD-6'SLN vs CD-6SLN; ml vs mL, etc.

Reviewer 3 Report

This paper describes the influenza inhibition and virucidal activity of CD-6’SLN.  The synthesis of CD-6’SLN is not acceptable because of poor structural elucidation.  The author should add further structural information and show the purity of the compound.  Other biological evaluations were acceptable.  These methodologies can be used by organic chemists and biochemists  This reviewer, therefore, would like to recommend this manuscript for publication in Pharmaceuticals with minor revision.

Structural elucidation of CD-6’SLN is needed.

The author should cite “Furuike, T, Sadamoto, R, Niikura, K, Monde, K, Sakairi, N, Nishimura, SI, Tetrahedron 61, pp. 737-1742, DOI: 10.1016/j.tet.2004.12.035.
